# Meat Juice and Oral Fluid as Alternatives to Serum for Aujeszky Disease Monitoring in Pigs

**DOI:** 10.3390/microorganisms11102418

**Published:** 2023-09-27

**Authors:** Emanuele Carella, Claudio Caruso, Ana Moreno, Alessia Di Blasio, Francesca Oberto, Nicoletta Vitale, Loretta Masoero

**Affiliations:** 1Istituto Zooprofilattico Sperimentale del Piemonte, Liguria e Valle d’ Aosta, Via Bologna 148, 10154 Torino, Italy; francesca.oberto@izsto.it (F.O.); loretta.masoero@izsto.it (L.M.); 2Azienda Sanitaria Locale CN1 (ASL-CN1), Via Pier Carlo Boggio 12, 12100 Cuneo, Italy; claudio.caruso@aslcn1.it; 3National Reference Centre for Aujeszky Disease, Istituto Zooprofilattico Sperimentale della Lombardia e dell’Emilia Romagna, Via Bianchi 7/9, 25124 Brescia, Italy; anamaria.morenomartin@izsler.it; 4Azienda Sanitaria Locale TO3 (ASL-TO3), Via Poirino 9, 10064 Pinerolo, Italy; alessia.diblasio@aslto3.piemonte.it

**Keywords:** meat juice, oral fluid, Aujeszky Disease Virus, ELISA, pigs

## Abstract

Aujeszky Disease Virus (ADV) is a double-stranded DNA virus with a lipoprotein envelope. The natural hosts of the infection are Suidae, but the virus can infect many other mammals. The gold-standard method identified by the WOAH for the diagnosis of Aujeszky disease is the ELISA method. The objective of this study was to compare the performance of meat juice and oral fluid matrices using a commercial ELISA kit designed for serum. A total of 80 blood and oral fluid samples were collected from four pig farms selected for this study. Diaphragm muscle samples of about 100 g and blood samples were collected from 213 animals at the abattoir. These biological matrices were collected from the same animals and tested using a competitive ELISA kit to detect antibodies against ADV. The relative accuracy of the meat juice compared to that of the serum was 96.7% (95% CI: 93.3–98.7%), with 206 correct results out of 213. The relative accuracy of the oral fluid compared to that of the serum was 61.3% (95% CI: 49.7–71.9%), with 58 correct results out of 80. Meat juice has a better combination of sensitivity and specificity than oral fluid. The usage of meat juice in routine diagnostic examinations could be achieved after further investigations to optimize the procedure.

## 1. Introduction

Aujeszky Disease Virus (ADV), which belongs to the family Herpesviridae and subfamily Alphaherpesvirinae, is a double-stranded DNA virus with a lipoprotein envelope, often referred to as Pseudorabies Virus (PRV) [1]. The natural hosts of the infection are Suidae, but the virus can infect many other mammals, including ruminants, carnivores, and rodents [1,2]. These species are considered dead-end hosts because they develop nervous symptoms that lead to a fatal outcome in a short time but cannot spread the virus in the environment. In these species, the symptomatology is characterized by an uncontrollable itching caused by acute neuropathy, which is not present in suids [3]. In pig farms, the virus can spread rapidly with very high morbidity. Symptoms vary according to the age of the specimens; piglets can develop highly lethal neurological symptoms, while finishers can develop respiratory symptoms with hyperthermia, dyspnea, and cough. In sows, Aujeszky Disease (AD) causes abortions and infertility accompanied by periorchitis [4]. Infected pigs shed the virus two days after infection, and the respiratory system is the main gateway for the virus into the body. In addition, milk, semen, vaginal secretions, feces, and occasionally urine are also infective. Transmission can also occur via direct or indirect contact through infected water, feed, and fomites [1]. The virus can remain latent in the trigeminal ganglia for long periods and cause recrudescence after a latency period, during which pigs appear healthy [5]. Therefore, AD has a considerable economic impact in the pig industry, mainly causing a decrease in production and a trade restriction, resulting in economic loss [6].

The “DIVA” (Differentiating Infected from Vaccinated Animals) strategy based on the use of deleted marker vaccines (gE-) and specific discriminatory serological tests (competitive gE ELISA) represents the key control element for AD. It is the gold-standard method, identified by the World Organization for Animal Health (WOAH), for the diagnosis of AD is serum neutralization [7]. However, it has currently been replaced by ELISA methods, which are more versatile and economical, used in the surveillance plans of the “DIVA” strategy [8]. Competitive ELISA tests represent the only routine tests available for the discrimination of infected animals from those vaccinated. It has been observed that glycoprotein E of ADV represents an ideal target; it is not essential to viral replication but capable of inducing a long-lasting humoral response in immunized animals [9]. The deletion of gE glycoprotein in the vaccine and the detection of antibodies against this immunoglobulin enables the differentiation of infected animals from vaccinated ones [10,11].

Adapting existing serological protocols to new matrices, such as meat juice and oral fluid, could facilitate the serological surveillance of various pathogens and reduce sampling costs. Meat juice has previously been used in wild boar and pigs as a biological matrix for detecting IgM and IgG antibodies against antigens of *Salmonella*, porcine reproductive and respiratory disease virus, Aujeszky’s disease virus, *Toxoplasma gondii*, and Hepatitis E virus [12,13,14,15,16,17,18,19,20,21,22]. Sample collection at abattoirs for serological screening is advantageous compared to sample collection at the farm as it avoids handling live animals and ensures the safety of the veterinarian and technician who collects the samples [16]. Moreover, muscle sample collection at abattoirs does not impact animal care and welfare [23,24]. Indeed, it can be easily performed at various stages of the slaughtering chain, enabling the monitoring of disease serological status at the end of the pig production cycle [16,18,19].

The use of oral fluid for assessing herd health status in swine has been proposed as a low-cost approach to disease surveillance, including AD, swine vesicular disease (SVD), and porcine reproductive and respiratory syndrome (PRRS) [25,26,27]. Oral fluid has been shown to contain various biomolecules, etiological agents, and IgG, IgM, and IgA antibodies typically detected in blood [28]. However, one limitation of these biological matrices is the lower concentration of IgM and IgG antibodies compared to serum samples [16,25].

The objective of this study was to evaluate the performance of a commercial ELISA kit designed for serum on meat juice and oral fluid matrices from pigs. Specifically, the presence of specific antibodies against the ADV in domestic pigs was evaluated using alternative biological matrices and comparing their results to those obtained with serum.

## 2. Materials and Methods

### 2.1. Study Design for the Evaluation of ELISA Kit Performance Using Meat Juice

Four pig farms were selected for the collection of biological matrices. One of these farms had animals that tested positive for Aujeszky’s disease and was considered infected because samples collected by a veterinary official, as part of the Control and Eradication Program prior to the study, tested positive. The pigs of the remaining three farms, which were free from ADV, were vaccinated based on the “DIVA” strategy, in accordance with the requirements of the regional control and eradication program implemented in 2018.

A total of 213 sera and diaphragm muscles were collected from the same pigs (study group A) reared in four different farms located in Piedmont (Italy), in 2018, to evaluate the performance of the competitive ELISA kit. The pigs from study group A were deceased, and sampling was performed at the abattoir. This sample size enabled a proportion of agreement higher than 96% to be estimated with up to 8 errors, at a 95% confidence level. With regard to the evaluation of relative specificity, 163 pigs from the three farms that had been ADV-free for at least three years were selected. With a 95% confidence level, this sample size enabled an expected specificity of the meat juice compared to the serum at 99.9% to be evaluated, with the lower limit of the range equal to 94%. Finally, 50 pigs from the ADV-infected farm were selected for sensitivity evaluation. The final sample size of 50 pigs enabled an expected sensitivity of 98% to be evaluated, with the lower limit of the range equal to 89%, at a 95% confidence level.

### 2.2. Study Design for the Evaluation of ELISA Kit Performance Using Oral Fluid

A total of 80 sera and oral fluid samples were collected from pigs (study group B) reared in four different farms located in Piedmont, Italy, in 2018, to evaluate the performance of the competitive ELISA kit. The number of pigs sourced from the three farms that had been ADV-free for at least three years was 28, while the remaining 52 came from the farm that had tested positive for the presence of ADV. The animals from study group B were alive, and sampling was performed at the farm. This was due to the poor yield of any oral fluid collected at the abattoir from stunned subjects. The sample size of 80 animals allowed us to estimate an expected accuracy of 95%, with the lower limit of the range equal to 88%, with up to 4 errors at a 95% confidence level. The 28 pigs from the three ADV-free farms were selected to evaluate the specificity. The selection allowed us to determine the specificity of the lower limit of the range equal to 87% and the specificity of oral fluid at 99.9%, at a 95% confidence level. Finally, 52 pigs from the ADV-infected farm were selected for sensitivity assessment. This sample size allowed us to estimate the sensitivity of the lower limit of the range equal to 90% for an expected specificity of 98%, at a 95% confidence level.

### 2.3. Sample Collection

A total of 80 blood and oral fluid samples were collected from the herds of the four pig farms. Oral fluid was collected in test tubes after animal containment by placing an oral swab near the buccal cavity and exploiting spontaneous salivation. Blood was collected using special test tubes without anticoagulants to compare the performance of oral fluid using a commercial ELISA kit designed for serum. Each swab was placed in a special V-bottom test tube and stored at 4 °C. These swabs enabled the collection of 200–700 µL of oral fluid.

Diaphragm muscle samples of about 100 g were removed from 213 animals prior to refrigeration and placed in a sterile disposable plastic bag. Blood was also collected during slaughter at the abattoir using special test tubes without anticoagulant to compare the performance of meat juice using a commercial ELISA kit designed for serum. After collection at the abattoir, the diaphragm samples were frozen at −20 °C then placed in a refrigerator at 4 °C for 20 h, before being stored at room temperature for approximately 2 h to facilitate thawing. After the freeze/thaw cycle, meat juice was collected directly from the sterile disposable plastic bag without centrifugation and stored at −20 °C. The volume of meat juice obtained from 100 g of diaphragm muscle varied from 300 µL to 2 mL. The meat juice samples were thermally treated at 56 °C for 30 min prior to serological analysis. All the animals tested from study A and study B were grow–finish pigs.

### 2.4. Serological Analysis of Oral Fluid

Blood and oral fluid samples from the same animals were tested using a competitive ELISA kit for the detection of specific antibodies against ADV glycoprotein E, which enables the discrimination between infected and vaccinated animals (IDEXX PRV/ADV gI Ab test, Westbrook, ME, USA). According to the manufacturer’s protocol, the samples were centrifuged at 1650× *g* for 15 min, and the supernatant obtained was then incubated for 20 h at 4 °C to ensure better sensitivity. Both sera and oral fluids from the same animal were tested using the same ELISA kit (IDEXX PRV/ADV gI Ab test), adopting a primary dilution of 1:2. The washing procedure and incubation with the secondary antibody were performed the following day, as recommended by the manufacturer. The plate was finally read at a wavelength of 650 nm. The percent inhibition (S/N) was calculated by dividing the absorbance of each sample by the mean negative control plate absorbance. According to the manufacturer, samples with an S/N value lower than 0.60 were classified as positive, samples with an S/N higher than 0.70 were classified as negative, and samples with an S/N between 0.60 and 0.70 were considered doubtful. Serum and oral fluids that were tested as doubtful were subjected to a second test session.

### 2.5. Serological Analysis of Meat Juice

The same competitive ELISA kit (IDEXX PRV/ADV gI Ab Test) used for oral fluids was employed to detect antibodies against the gE of ADV in the meat juice. The immunoassay test was performed by testing both the serum and meat juice obtained from the same animal. Undiluted meat juice samples (200 µL per well) were analyzed using the ELISA kit, while the respective sera were tested at a primary dilution of 1:2, according to the manufacturer’s instructions. Plate washing, incubation with the secondary antibody, development, and the interpretation of results were carried out according to the manufacturer’s instructions for the serum matrix. The plate was read at a wavelength of 650 nm. Percent inhibition (S/N) was calculated by dividing the absorbance of each sample by the mean negative control plate absorbance. Serum and meat juice samples that tested as doubtful were subjected to a second test session.

### 2.6. Statistical Analysis

The accuracy of meat juice was evaluated using the optical density (OD) obtained from study group A, whereas the accuracy of oral fluid was evaluated using the OD obtained from study group B. The following indices were used to estimate accuracy: the sensitivity, the specificity, the proportion of false positives, the proportion of false negatives, and the Youden index. The Youden index was calculated to assess the diagnostic efficacy of the ELISA test. It represents the maximization of the sum of sensitivity and specificity for a threshold value (cut-off), which correctly distinguishes between positive and negative samples. The Youden index values range from 0 to 1, where a value of 0 corresponds to a completely ineffective diagnostic test, while a value of 1 represents a perfectly effective diagnostic test [29]. The proportion of positives was calculated for each matrix. The binomial distribution was used to calculate the exact confidence limit of each proportion. The Cohen’s Kappa index was used to evaluate the agreement between the biological matrices [30]. The Kappa index indicates the proportion of agreement, excluding that expected by chance, for categorical variables. Kappa values near 1 indicate perfect agreement, while a Kappa value of 0 indicates that all agreement is due to chance. According to Landis and Koch [31], a Kappa index value between 0.60 and 0.8 is considered good agreement. All statistical analyses were performed using SAS System 9.4.

## 3. Results

### 3.1. Comparison of ELISA Kit Results Using Meat Juice and Sera, and Evaluation of Their Performance

A total of 213 pigs were sampled to evaluate the performance of the ELISA kit using serum. All 50 sera from the ADV-infected farm tested positive, while all 163 sera from the three ADV-free farms tested negative. The optical density distribution of all serum samples is shown in Figure 1A. The diagnostic accuracy of the serum for herd health was 100% (95% CI: 98.6–100%), with 215 correct results out of 215 (Table 1). The diagnostic specificity was 100% (95% CI: 98.3–100%), with 163 negative samples out of 163 specimens from the three ADV-free pig farms. The diagnostic sensitivity was 100% (95% CI: 93.2–100%), with 52 positive animals out of 52 infected animals. The Youden and Kappa indexes were 1 (95% CI: 96.7–100%), suggesting “perfect” test performance.

With regard to the analysis of the meat juice, all 50 samples from the ADV-infected farm tested positive using the same cut-off suggested for serum by the ELISA kit manufacturer. In contrast, analysis of the 163 meat juices of the three ADV-free farms detected 156 samples as properly negative and 7 as false positives. The optical density distribution of all meat juice samples is shown in Figure 1B. The diagnostic accuracy of the meat juice for herd health was 96.7% (95% CI: 93.3–98.7%), with 206 correct results out of 213 (Table 1). The diagnostic specificity was 95.7% (95% CI: 91.3–98.3%), with 156 samples testing negative out of the 163 and 7 false positives, while the diagnostic sensitivity was 100% (95% CI: 92.9–100%), with 50 positive animals out of 50. The relative accuracy of the meat juice compared to the serum was 96.7% (95% CI: 93.3–98.7%), with 206 correct results out of 213. The relative specificity was 95.7% (95% CI: 91.3–98.3%), with 156 animals testing negative for meat juice compared to 163 animals testing positive for serum (Table 1). The relative sensitivity was 100% (95% CI: 92.9–100%), with 50 positive meat juice out of 50 positive sera. The Youden index was 0.957 (Table 1), while the Kappa index was 0.913 (95% CI: 0.849–0.976), suggesting a good test performance.

### 3.2. Comparison of ELISA Kit Results Using Oral Fluid and Sera, and Evaluation of Their Performance

A total of 80 pigs were sampled to evaluate the performance of the ELISA kit using serum. All 52 sera from the ADV-infected farm tested positive except for one specimen, which also tested negative for oral fluid. In contrast, all 28 sera from the three ADV-free farms tested negative. The optical density distribution of all sera samples is shown in Figure 1C. The diagnostic accuracy of the serum matrix for herd health was 98.8% (95% CI: 93.2–99.9%), with 79 correct results out of 80 (Table 2). The diagnostic specificity was 100% (95% CI: 87.7–100%), with 28 negative animals out of 28. The diagnostic sensitivity was 98.1% (95% CI: 89.7–99.9%), with 51 positive animals out of 52. The Youden index was 98.1%, while the Kappa index was 0.973 (95% CI: 92–100%), suggesting “almost perfect” test performance.

With regard to the analysis of the oral fluids, out of the 52 samples from the ADV-infected farm, 30 tested positive, and 22 tested negative using the same cut-off suggested for serum by the ELISA kit manufacturer. In contrast, all 28 sera from the three ADV-free farms tested negative. The optical density distribution of all oral fluid samples is shown in Figure 1D. The diagnostic accuracy of the oral fluid for the herd health was 60% (95% CI: 48.4–70.8%), with 58 correct results out of 80 (Table 2). The diagnostic specificity was 100% (95% CI: 87.7–100%), with 28 negative animals out of 28. The diagnostic sensitivity was 38.4% (95% CI: 25.3–52.3%), with 30 positive animals out of 52 infected animals. The relative accuracy of the oral fluid compared to the serum was 61.3% (95% CI: 49.7–71.9%), with 58 correct results out of 80. The relative specificity was 100% (95% CI: 87.7–100%), as 28 samples tested negative for both oral fluids and sera (Table 2). In contrast, the relative sensitivity was 39.2% (95% CI: 25.8–53.9%), with 30 positive oral fluids compared to 51 positive sera. The Youden index was 0.384 (Table 2), while the Kappa index was 0.304 (95% CI: 0.170–0.438), suggesting a weak degree of agreement.

## 4. Discussion

The WOAH manual indicates that serum is the optimal matrix for the detection of antibodies against Aujeszky’s disease virus [7]. The diagnostic accuracy, the diagnostic sensitivity and specificity, the Youden index, and the Kappa index confirm that serum is the standard matrix for the commercial ELISA kits routinely used in the eradication plans for Aujeszky’s disease. However, this biological matrix presents some disadvantages, such as the identification of the animal to be examined, the difficulty of performing individual sampling on a large scale, the impact of sampling on animal welfare, and the presence of a qualified veterinary staff in the pig farm to collect the serum [16]. Therefore, the use of alternative matrices has been investigated by adapting them to the commercial ELISA kit already available for serum. These new matrices have been selected thanks to the ease of sampling, the absence of effect on the commercial value of the carcasses, and the possibility of sampling in a systematic manner. The meat juice, obtained by freezing and thawing a muscle sample, has been reported for the detection of several pathogens [16,17,18]. Furthermore, oral fluid has also been used as an alternative matrix for monitoring different diseases, as it contains the pathogens responsible for the infection and their respective antibodies [15,25]. However, Wallander et al. [19] have shown that ELISA, using meat juice, might not be reliable in detecting animals with low-grade infections and/or low antibody levels. In fact, this alternative matrix shows lower levels of specific antibodies than serum, which shows an anti-gE antibody titer ten times greater than that of corresponding meat juice [12,17]. Nevertheless, it has been suggested that the lower sensitivity of meat juice serology could be improved by adjusting its dilution according to its blood content [32]. Moreover, this biological matrix presents a high concentration of several proteins compared with the serum samples. These proteins may interfere with the anti-gE serum used in the kit, likely through a process of non-specific saturation [16]. Therefore, in our study, the meat juice samples were thermally treated at 56 °C for 30 min to avoid any non-specific reactions that may occur during serological test analyses.

The standardization of the serological method is important to ensure that the results are accurate and consistent. The incubation temperature and time were observed to have a considerable influence on the sensitivity of an ELISA [33,34]. It has been shown that a difference of only 0.5 °C is sufficient to obtain a different qualitative result. Consequently, if the incubation temperature is not defined rigorously, this can lead to contradictory results that cannot be compared with one another [34]. Furthermore, careful consideration should be given to the role of sample dilution and cut-off values when aiming to standardize the sample analysis.

The volume of exudate obtained post-thaw and the concentration of antibodies can vary significantly depending on the process used to extract the meat juice and the muscle type collected. Therefore, the standardization of both the collection procedure and the choice of muscle type is crucial to ensure the consistency and comparability of analytical results. The concentrations of antibodies in meat juice are related to the different degrees of vascularization in the muscles chosen for sampling [12,19,21]. Heart samples have significantly higher levels of antibodies, compared with other muscles, and the greatest variation [19]. The diaphragm and the tongue generally have intermediate levels, but their antibody concentrations are steadier. In fact, the highest concentrations of antibodies have been detected in meat juice from the heart, followed by the diaphragm, tongue and semitendinosus [19]. Consequently, it may be concluded that the heart is the most appropriate muscle for serological analysis compared to the diaphragm. However, it should be noted that the levels of antibodies present in the heart show significant variations in concentration, which may impact the results of the analysis. Moreover, in many countries, the heart is considered a delicacy and, therefore, not suitable for extracting meat juice. Therefore, the diaphragm muscles could be considered the best biological matrix via which to obtain the meat juice. In fact, Le Potier et al. [16] suggested that the diaphragm is the best muscle to obtain a good volume of meat juice from due to its accessibility and low commercial value. Moreover, the diaphragm is already collected at the abattoir for the detection of *trichinella* in Italy. Therefore, there is no need for further sampling, but it could be used both to evaluate the parasite’s presence and to obtain meat juice. The diaphragm produced sufficient muscle exudate for the analyses and was, therefore, used in this study. In fact, a volume of meat juice ranging from 300 µL to 2 mL was obtained from our muscle samples, which is more than enough to perform the analysis.

Several studies have investigated the utilization of various ELISA techniques for the employment of meat juice as an alternative biological matrix to serum, yielding disparate results. Le Potier et al. [16] performed a comparative serological analysis of 389 pairs of samples of serum and meat juice obtained from different pigs in order to detect anti-gE antibodies against Aujeszky’s disease virus. It was reported that the sensitivity in the meat juice was 93.2% (95% CI: 88.8–96.9%), while the specificity was 98.3% (95 CI: 96.5–99.5%) if the doubtful and negative results were grouped together. Instead, when the doubtful results were grouped with the positive results, the sensitivity was 98.1%, and the specificity was 93%. It has also been shown that the results correlated well with those obtained using serum samples, since the coefficient of concordance was 93.3%. De Lange et al. [18] conducted a blind study analyzing meat juice samples using three commercially available ELISA-gE kits for Aujeszky’s disease serum analysis. A preliminary sensitivity study was performed on 45 samples from seropositive sows. For the specificity study, 1879 samples were analyzed, including 1423 from finishers and 456 from sows. The sensitivities of the kits, calculated using doubtful results considered positive, ranged from 80% to 91%. Conversely, when doubtful results were considered negative, the sensitivities of the kits ranged from 73% to 80%. The specificity of the kits was over 0.995 when the doubtful results were considered positive, and over 0.999 when doubtful results were considered negative. Nielsen et al. [12] demonstrated that meat juice can be used as an alternative to serum for the serologic detection of specific *Salmonella* antibodies by using ELISA. Ranucci et al. [21] showed a substantial concordance between serum and meat juice for the detection of antibodies against *Toxoplasma gondii*. The results described above are consistent with our study, in which the meat juice collected at abattoirs showed high sensitivity (100%) and specificity (95.7%). The degree of agreement between this alternative matrix and serum was good, as demonstrated by both the Youden index (0.957) and the Kappa index (0.913). However, Vico and Mainar-Jaime [13] observed a significant discrepancy between the results of ELISA tests performed on serum and meat juice matrices. They suggested that the choice of matrix for performing ELISA to detect the prevalence of *Salmonella* in swine should be carefully considered. The overall correlation coefficient between the serum and meat juice results was low (0.53). Furthermore, other studies have determined that the use of ELISA tests with meat juice as a matrix is ineffective. The primary issue was the high rate of false negative samples due to the low blood content in the tested meat juice [19,35,36]. As a result, meat juice may not be a suitable alternative matrix to serum for ensuring the absence of ADV. However, it could be applied for monitoring purposes, particularly when blood collection at the pig farm is difficult.

In veterinary medicine, oral fluids have been utilized for the detection of *Escherichia coli* O157:H7, *Salmonella* in feedlot cattle, and feline leukemia virus in cats. In swine, specific antibodies against porcine reproductive and respiratory syndrome virus (PRRSV) and porcine circovirus type 2 (PCV2) have been detected in oral fluid following experimental inoculation [15]. At the pen level, 77% of the PRRSV qRT-PCR oral fluid and serum results agreed, indicating that PRRSV is detectable in the oral fluid matrix and could be suitable for monitoring PRRSV circulation in a herd. Additionally, PCV2 has been detected in oral fluids, suggesting that this alternative matrix could be used to monitor its circulation within a group of animals [15]. In our study, oral fluid was found to be unsuitable for the analysis of Aujeszky’s disease. The diagnostic specificity was 100%, whereas the diagnostic sensitivity was very low (38.4%). The degree of agreement between serum and oral fluid was weak, as demonstrated by both the Youden index and the Kappa index. Nevertheless, the sensitivity result was better than that obtained by Panyasing et al. [25], who used the same ELISA kit to detect antibodies against ADV. These results suggest that oral fluid does not present problems relating to non-specific reactions and shows high specificity. However, the issue of the incorrect identification of positive samples could render this biological matrix unfit as an alternative to serum for the detection of antibodies against ADV.

## 5. Conclusions

This study provides evidence for the feasibility of using meat juice as an alternative matrix for antibody detection in serological assays, which could be particularly useful in situations in which blood collection is difficult to perform. Furthermore, the use of meat juice could provide a non-invasive method for monitoring antibody levels in pigs, which could help improve the control and prevention of Aujeszky’s disease. However, further investigation is needed to optimize the serological procedure for using alternative matrices in routine diagnostic examinations, such as the role of incubation temperature and time, sample dilution conditions, and cut-off values.

With regard to oral fluids, these exhibit lower accuracy and sensitivity values compared to serum. This finding suggests a reduced ability to detect infection and a probable underestimation of the actual seroprevalence when ELISA is performed using this alternative matrix.

Meat juice appears to have a better combination of sensitivity and specificity compared to oral fluid, and its usage in routine diagnostic examinations could be achieved after further investigations to optimize the procedure.

## Figures and Tables

**Figure 1 microorganisms-11-02418-f001:**
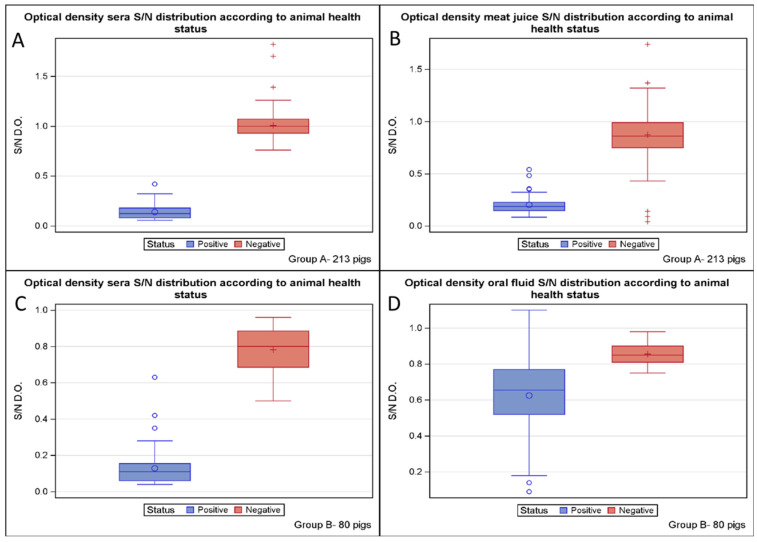
The optical density distribution of sera (**A**) and meat juice (**B**) according to the animal health status is shown for group study A of 213 pigs on the top, while the optical density distribution of sera (**C**) and oral fluid (**D**) according to the animal health status is shown for group study B of 80 pigs on the bottom.

**Table 1 microorganisms-11-02418-t001:** Accuracy of ELISA gE on sera and meat juice evaluated on 213 animals. N° of animals tested (N), true positive (TP), true negative (TN), false positive (FP), % false positive (%FP), false negative (FN), % false negative (%FN), sensitivity and confidence interval 95% (SE CI95%), specificity and confidence interval 95% (SP CI95%) Youden’s J.

ELISA gE	N	TP	TN	FP	%FP	FN	%FN	SE (CI95%)	SP (CI95%)	Youden’s J
Sera	213	50	163	0	0%	0	0%	100% (92.9–100%)	100% (98.3–100%)	100%
Meat Juice	213	50	156	7	4.29%	0	0%	100% (92.9–100%)	95.7% (91.3–98.3%)	95.7%

**Table 2 microorganisms-11-02418-t002:** Accuracy of ELISA gE on sera and oral fluid evaluated for 80 animals. N° of animals tested (N), true positive (TP), true negative (TN), false positive (FP), % false positive (%FP), false negative (FN), % false negative (%FN), sensitivity and confidence interval 95% (SE CI95%), specificity and confidence interval 95% (SP CI95%) Youden’s J.

ELISA gE	N	TP	TN	FP	%FP	FN	%FN	SE (CI95%)	SP (CI95%)	Youden’s J
Sera	80	51	28	0	0	1	1.9%	98.1% (89.7–99.9%)	100% (87.7–100%)	98.1%
Oral fluids	80	20	28	0	0	32	61.5%	38.4% (25.3–52.3%)	100% (87.7–100%)	38.4%

## Data Availability

The data presented in this study are available upon request from the corresponding authors.

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
