# Peer review of "Meat Juice and Oral Fluid as Alternatives to Serum for Aujeszky Disease Monitoring in Pigs"

_microorganisms, 2023, doi:10.3390/microorganisms11102418_

Round 1

Reviewer 1 Report

In this study, a total of 80 blood and oral fluid samples were collected from four pig farms selected for this study. Diaphragm muscle samples (100 g each) and blood samples were collected from 213 animals at the abattoir. These biological matrices were tested using a competitive ELISA kit to detect antibodies against gE protein of ADV. The relative accuracy of the meat juice compared to that of the serum was 96.7%. The relative accuracy of the oral fluid compared to that of the serum was 61.3%. Meat juice has a better combination of sensitivity and specificity than oral fluid. The study is helpful for evaluation of diaphragm muscle samples as alternative matrices to detect antibodies against ADV. However, there are some comments on this paper.

1.     gE antibody titration is usually to find Infected animals for ADV eradication program, while diaphragm muscle samples could only be collected at the abattoir or at autopsy stage. The value of diaphragm muscle samples as alternative matrices to detect antibodies against ADV is limited. The authors are encouraged to discuss about this concern.

2.     I do not understand that why oral fluid samples have not been included in study group A, since the number of pigs employed in this study is more than that of those involved in study group B,and the comparison results obtained in the former seemed to be more robust than the latter. The authors need to discuss about the study design.

3.     Either for study group A or for group B, the information of tested pigs should be detailed. Are pigs grow-finish pigs, sows and/or boars?

4.     What are the criteria to define “ADV infected farm” in study group A or “farm that tested positive for the presence of ADV” in study group B? The criteria should be detailed.

5.     In table 1, there are 7 diaphragm muscle samples defined as false positive, are these 7 samples confirmed to be true negative using virus neutralization assay?

6.     The table note “true positive (TP), true negative (TN)” seemed to be missed in both table 1 and 2.

The quality of English language is acceptable.

Reviewer 2 Report

I do not have any significant comments on this manuscript. From the point of view of potential conflict of interest, it will be good to explain the choice of commercial kit for this study. 

By the information from IDEXX PRV/ADV gI Ab Test: Pseudorabies Virus/Aujeszky's Disease (PRV) - IDEXX US

several IDEXX PRV/ADV gI Ab test available for purchase, do not know that is the differ between this kits, but for reproducibility of results better to clarify this point.

It the References authors mentioned paper from Le Potier, M.F.; Fournier, A.; Houdayer, C.; Hutet, E.; Auvigne, V.; Hery, D.; Sanaa, M.; Toma, B. Use of muscle exudates for the 414 detection of anti-gE antibodies to Aujeszky’s disease virus. Vet. Rec. 1998, 143, 385-387. doi: 10.1136/vr.143.14.385 and other publications on the same topic.

From this point not clear what new authors add to the idea of using of meat juice as alternative to serum for Aujeszky disease monitoring in pigs?

English is fine for me

Round 2

Reviewer 1 Report

All my comments have been placed down, the paper can be accepted.